# Hierarchical Cluster Analysis Based on Clinical and Neuropsychological Symptoms Reveals Distinct Subgroups in Fibromyalgia: A Population-Based Cohort Study

**DOI:** 10.3390/biomedicines11102867

**Published:** 2023-10-23

**Authors:** Sara Maurel, Lydia Giménez-Llort, Jose Alegre-Martin, Jesús Castro-Marrero

**Affiliations:** 1Department of Medicine, Universitat Autònoma de Barcelona, 08193 Barcelona, Spain; saranieves.maurel@uab.cat; 2Department of Psychiatry and Forensic Medicine, School of Medicine, Universitat Autònoma de Barcelona, 08193 Barcelona, Spain; lidia.gimenez@uab.cat; 3Institut de Neurosciències, Universitat Autònoma de Barcelona, 08193 Barcelona, Spain; 4Division of Rheumatology, Clinical Unit in ME/CFS and Long COVID, Vall d’Hebron University Hospital, Universitat Autònoma de Barcelona, 08035 Barcelona, Spain; jose.alegre@vallhebron.cat; 5Division of Rheumatology, Research Unit in ME/CFS and Long COVID, Vall d’Hebron Research Institute, Universitat Autònoma de Barcelona, 08035 Barcelona, Spain

**Keywords:** chronic pain, fibromyalgia, cluster analysis, neuropsychological symptoms, fatigue, mindfulness

## Abstract

Fibromyalgia (FM) is a condition characterized by musculoskeletal pain and multiple comorbidities. Our study aimed to identify four clusters of FM patients according to their core clinical symptoms and neuropsychological comorbidities to identify possible therapeutic targets in the condition. We performed a population-based cohort study on 251 adult FM patients referred to primary care according to the 2010 ACR case criteria. Patients were aggregated in clusters by a K-medians hierarchical cluster analysis based on physical and emotional symptoms and neuropsychological variables. Four different clusters were identified in the FM population. Global cluster analysis reported a four-cluster profile (cluster 1: pain, fatigue, poorer sleep quality, stiffness, anxiety/depression and disability at work; cluster 2: injustice, catastrophizing, positive affect and negative affect; cluster 3: mindfulness and acceptance; and cluster 4: surrender). The second analysis on clinical symptoms revealed three distinct subgroups (cluster 1: fatigue, poorer sleep quality, stiffness and difficulties at work; cluster 2: pain; and cluster 3: anxiety and depression). The third analysis of neuropsychological variables provided two opposed subgroups (cluster 1: those with high scores in surrender, injustice, catastrophizing and negative affect, and cluster 2: those with high scores in acceptance, positive affect and mindfulness). These empirical results support models that assume an interaction between neurobiological, psychological and social factors beyond the classical biomedical model. A detailed assessment of such risk and protective factors is critical to differentiate FM subtypes, allowing for further identification of their specific needs and designing tailored personalized therapeutic interventions.

## 1. Introduction

Fibromyalgia (FM) is a chronic complex condition of unknown etiology, considered multifactorial and combining genetic and epigenetic factors that condition a persistent alteration in pain regulation mechanisms [1]. The diagnostic case criteria of the 2010 ACR identify it by the presence of widespread and diffuse musculoskeletal pain lasting more than 3 months and with painful tenderness to palpation in at least 11 of 18 tender points, defined as trigger points [2]. In FM, pain is the most frequent and disabling symptom; it is considered as a chronic pain syndrome that presents with neurophysiological alterations similar to other chronic pain states but is also accompanied by other functionally limiting clinical symptoms. This multiplicity of other symptoms includes fatigue, sleep disturbances, irritable bowel syndrome, paresthesia, concentration and memory problems, muscle stiffness, mood disturbances, high comorbidity of anxiety/depression symptoms and somatoform disorders [3,4,5]. 

Fibromyalgia is one of the leading health problems worldwide due to its high prevalence causing significant clinical and social burden, which may further exacerbate the patients’ level of disability [6], chronic pain [7] and reduction in the quality of life that it entails, as well as the high healthcare expenditure that it generates [8,9]. Furthermore, this debilitating condition can start and profoundly affect the psychology and life of people during their life cycle [10,11]. In many countries, such as the U.S., Canada, Norway, Sweden and Brazil, FM is already recognized as a disease with high rates of disability.

The prevalence of FM in Spain according to the EPISER study is high and has been estimated as approximately 2.4% of the general population over 20 years of age, representing approximately 700,000 people affected [12,13]. It also presents a clear gender difference, being predominantly in women (4.2% in women, compared to 0.2% in men) with a ratio of 1:21. Although FM can manifest itself at all ages, it presents a maximum peak between 40 and 50 years of age (4.9%). Geographically, the highest prevalence is found in Europe, and Spain is one of the European countries with the highest prevalence after Germany, Portugal, Italy and Turkey.

It is important to note that heterogeneity and variability have been found in people with FM, both in clinical symptoms [14] and in differences in psychological processing [12,15], altered cardiovascular reactivity and distorted pain perception [15,16,17]. This heterogeneity and variability in symptomatology, but also the difficulties in reaching a rigorous diagnosis, directly affect prevalence studies, resulting in 2–4% in the general population, 2–6% in primary care consultations and 10–20% in rheumatology consultations. In this regard, different subgroups of FM patients based on pressure–pain thresholds, psychological factors and coping strategies (anxiety/depression; catastrophism) have been defined, and personalized therapeutic approaches proposed. Their main aim is to be able to propose different strategies for coping with pain and to offer the FM patients personalized therapeutic approaches [18].

In the present work, our study aimed to identify subgroups of FM patients using a hierarchical cluster analysis based on clinical and neuropsychological variables. Thus, clinical variables such as pain, fatigue, sleep quality, stiffness, anxiety, depression and disability at work with physical and emotional impact were studied. The psychological variables explored were acceptance, mindfulness, positive affect and negative affect, catastrophizing, surrender and perceived injustice in the FM population.

## 2. Methods

### 2.1. Study Participants

We included 251 FM patients from the primary care health center of Aragon region, Spain (Zaragoza, Teruel and Huesca), and selected those patients who met the following inclusion criteria: patients diagnosed with FM according to the 2010 ACR case criteria [2], aged between 18–65 years, speaking and understanding Spanish, giving informed consent. Exclusion criteria included suffering from severe medical illness or severe neuropsychiatric disorder in Axis I, not collaborating or not signing the informed consent.

Patients were contacted by telephone to explain the characteristics of the study protocol. Subsequently, the patients were randomly summoned by a psychologist in postgraduate training in order to carry out the clinical interview and complete the protocols in person. In the personal interview, the conditions and completion of questionnaires, study aims, research conditions and the confidentiality of the process and data were explained. After obtaining informed consent, the tests and interviews were administered, lasting approximately one and a half hours per person. 

The study followed the norms of the Helsinki Convention and its subsequent modifications and the Madrid Declaration of the World Psychiatric Association. Informed consent was requested from all patients before being included in the study and voluntary and informed participation was guaranteed. At any time, the patient could decline to participate or answer any questions he/she considered appropriate, as well as revoke the previously signed informed consent.

This study is part of a project financially supported by the Health National Institute Carlos III in Madrid, Spain (grant number PI09/90301) and the data collection took place between July 2011 and May 2014.

### 2.2. Measures

Sociodemographic variables such as age, sex, place of residence, marital status, educational level, cohabitation, employment status and comorbidity were recorded. A general survey was also performed on each participant.

### 2.3. Health Status Variables

Physical impact variables of FM: pain, fatigue, sleep quality, stiffness and difficulty at work. The Spanish consensus version of the Fibromyalgia Impact Questionnaire (FIQ) was used. The Spanish FIQ version was used in this study [19]. The FIQ is composed of 10 items classified on a four-point Likert scale (0–3) and quantifies functional capacity, health status, pain intensity, sleep disturbances, muscle stiffness, fatigue, anxiety/depression and personal perception. It is an instrument recommended by the Spanish Society of Rheumatology (SER) to assess both disability and the global and physical impact of FM. The first item, consisting of 9 sub-items, focuses on the patient’s ability to perform physical activities and can be used in isolation to assess the patient’s degree of disability. The next two require the patient to indicate the number of days in the previous week that he/she felt well and how many days in the last week he/she stopped working due to illness. The remaining seven (4 to 10) refer to the ability to work, pain, fatigue, morning tiredness, stiffness, anxiety and depression, all measured (from 0 to 10) using visual analog scales (VASs). Higher scores indicate a greater degree of impact of the disease, a score of 70 or more is considered severe and the Cronbach’s alpha was 0.80 [14,20]. 

Variables of Emotional Impact or Distress, Anxiety and Depression. The Hospital Anxiety and Depression Scale (HADS), a self-reported measure, was used to detect anxiety and depression in people with medical illnesses. It comprises 14 items that are scored on a 4-point Likert scale. It includes two subscales—anxiety (7 items) and depression (7 items)—which are scored independently; the higher the score, the greater the anxiety or depression. The HADS has been validated in Spanish. It is recommended by the SER as an instrument to assess the emotional state or impact of patients with FM (Rivera et al., 2006). The Spanish version has shown adequate test–retest reliability and a Cronbach’s alpha of 0.69 [21].

### 2.4. Neuropsychological Variables

Acceptance: It is measured with the Chronic Pain Acceptance Questionnaire (CPAQ). The Spanish validation of the questionnaire was used in this study. It measures pain acceptance as a predictor of well-being in patients with chronic pain. It is an abbreviated version of the CPCI-42 that maintains its psychometric properties. It analyzes 2 subscales: willingness to perform activities of daily living and acceptance of pain. It is composed of a list of 20 self-administered items, assessed on a rating scale from 0 (never true) to 6 (always true). The result of the two subscales is summed directly and the range is between 0 and 120. The higher the score, the greater the acceptance of pain and/or availability to perform the activities. The Spanish version has shown adequate test–retest reliability, internal consistency and construct validity [22].

Mindfulness: It is measured with the Mindful Attention Awareness Scale (MAAS) validated in Spanish. The mindfulness scale is composed of 15 items, which measure the construct of “being aware”, focused on the present and without judging the situations being experienced. The correctness of the items is from 1 (almost always) to 6 (almost never), summed directly. There is only one factor, which is the total of the scale. The range is between 15 (minimum mindfulness) and 90 (maximum mindfulness). In an attempt to control for socially desirable responses, participants are asked to respond according to what actually reflects their experience and not what they think it should be. Cognition, emotions, physical, interpersonal and general domains are assessed. The original MAAS had high internal consistency (Cronbach’s alpha is 0.85) [23].

Positive and Negative Affectivity (PANAS scales): Most studies on the structure of affect agree that affect is made up of two dimensions or factors: positive and negative affect. The PANAS scales have proven to be a valid and reliable measure to assess the presence and degree of positive and negative affect in clinical and normal populations of adolescents, adults and older adults, validated in Spanish. The PANAS is characterized by an internal consistency with alphas of 0.86 to 0.90 for positive affect and 0.84 to 0.87 for negative affect. The correlation between the two is invariably low, ranging from −0.12 to −0.23, which reinforces the idea that they are independent domains of affective well-being. Currently, this scale has an abbreviated version of 10 items that have proven to be cross-culturally reliable. PANAS consists of two mood scales with 10 items each for the assessment of positive and negative affectivity. The score for each range of the scale goes from 0 to 50. A positive affect is obtained by adding the odd-numbered items and negative affect by adding the even-numbered items. The scores for both affects are obtained by adding the numbers assigned to the 10 items that make up each of the two scales. The mean positive PANAS for a sample of young students was 32 and the mean negative PANAS was 23. People with scores above 38 for positive and below 16 for negative are characterized by an extremely positive affect balance. People with scores below 25 in positive and above 30 in negative give an extremely negative affect balance [24]. 

Catastrophizing: It is measured with the Spain-validated Pain Catastrophizing Scale (PCS). It consists of 13 items divided into three subscales that analyze rumination or meditation doubt (4 items), magnification or exaggeration (3 items) and hopelessness or helplessness (6 items), and its design focuses on feelings and thoughts related to pain, valuing them on a scale from 0 (not at all) to 4 (all the time). The 3 subscales are summed and the range is between 0 and 52. A higher score would correspond to a higher frequency and intensity of negative thinking and feelings regarding pain. The maximum total score is 52, such that a higher score would correspond to a higher frequency and intensity of negative thinking and feelings regarding pain [25].

Surrender: The Spanish version of the Pain Self Perception Scale was used. It is composed of 24 items that measure the self-processing of thoughts and feelings that can be experienced during an episode of severe pain. It is a self-administered questionnaire, where responses are measured using a 5-point scale between 0 (minimum intensity) and 4 (maximum intensity). There is only one factor, which is the global factor of the scale, and there are no subfactors. It is scored between 0 and 96 [26].

Psychological inflexibility: It is measured by the self-reported Psychological Inflexibility of Pain Scale, which contains 12 items, with two main subscales: avoidance (8 items) and cognitive fusion (4 items); both measure the inability to maintain our values in the presence of unpleasant thoughts, emotions and physical symptoms. The statements are scored on a Likert-type scale ranging from 1 (never true) to 7 (always true). Total 12-item scores can range from 12 to 84. Higher scores would indicate greater psychological inflexibility in the face of pain. Its psychometric properties are considered adequate [27].

Perceived injustice: This construct is measured by a questionnaire that reliably measures how a traumatic situation affects people’s lives validated in Spanish. It contains 12 items, with 2 subscales of severity/irreparability (6 items) and guilt/injustice (6 items) on a 5-point scale, from not at all (0) to all the time (4). The total scale scores range from 0 to 48. The psychometric properties of the injustice experiences questionnaire (IEQ) are considered adequate for use in the study [28,29].

### 2.5. Statistical Analysis

SPSS version 21 was used to perform cluster analysis, which is a multivariate technique that seeks to group variables to achieve maximum homogeneity in each group and the greatest difference between groups. We will rely on cumulative hierarchical algorithms, i.e., the method that forms groups by making larger and larger clusters. In our study, three cluster analyses were performed using the hierarchical clustering method (interval measure—Euclidean-by variables), as discussed below. First of all, we can say that the distance considered is the Euclidean distance (interval measure given by the following procedure, i.e., between each pair of elements, the difference in squared coordinates, the sum of all of them and finally their square root is considered). In addition, in order to compare the distances, the distances are standardized into Z-scores. The linkage technique is the inter-group average. 

Three cluster analyses were performed and the objective consisted first of all in grouping all the individuals into non-predefined subgroups and classifying or organizing them according to the 14 variables studied in order to obtain similar profiles with respect to psychological and clinical variables. From the result obtained, two other cluster analyses were performed, one with psychological variables (catastrophizing, surrender, mindfulness, acceptance, injustice, positive affect and negative affect) and another cluster analysis with respect to clinical variables (pain, fatigue, sleep quality, stiffness, anxiety, depression and disability at work).

## 3. Results

### 3.1. Participants’ Demographic and Clinical Characteristics

Table 1 shows the demographic and clinical features of the study population. In relation to sex, 96.1% were women and 3.9% were men. Participants had a mean age (± SD) of 52.4 ± 8.0 years (95% CI: 51.3–53.4, range: 3–70 years) and a mean length of time from onset of symptoms until inclusion of 18.3 ± 11.1 years. In all, 82.9% of the FM patients lived in Zaragoza and 11.2% in Teruel in Spain. Most (84%) had finished primary and secondary education. As for their living arrangements, most were married or living with a partner (73.7%) and almost half lived in their own home with their partner and children (47.4%). With regard to employment, 25.1% were working, 21.1% were permanently disabled and 12.3% were on sick leave.

### 3.2. Cluster Analysis of Clinical and Neuropsychological Parameters

Global cluster 1 analysis included pain, fatigue, sleep quality, stiffness, difficulty at work, anxiety/depression, surrender, injustice, mindfulness, catastrophizing, acceptance, positive affect and negative affect. These variables have been analyzed from a sample collected from 251 patients; 96.8% are valid cases and 3.2% are missing values. In our study, the matrix of distances between the different clinical and psychological variables mentioned above was performed and below we present the average (inter-group) linkage of the psychological constructs and clinical variables mentioned above (Table 2).

The dendrogram shows the formation of the clusters, as well as the distances between them. It can be seen, for example, that the variables (observations) closest to each other were pain (FIQ_5); fatigue (FIQ_6); sleep quality (FIQ_7); stiffness (FIQ_8); anxiety (Total HADS-A); depression (Total HADS-D); and difficulty at work (FIQ_4), which form the first group (distance closest to 0) and were the closest to each other. These are joined with injustice, catastrophizing (CATAST), negative affect (PANAS_negative) and positive affect (PANAS_positive). On the other hand, it was joined by proximity acceptance (CPAQ) and mindfulness and, finally, a fourth cluster with the variable surrender (Figure 1).

The summary of the subgroups obtained from the global analysis was as follows: cluster 1: FM patients with pain, fatigue, sleep quality, stiffness, anxiety, depression and difficulty at work; cluster 2: FM patients with injustice, catastrophizing, negative affect and positive affect; cluster 3: FM patients with mindfulness and acceptance; cluster 4: FM patients with surrender.

Cluster 2 analysis: pain, sleep quality, fatigue, stiffness, anxiety, depression and difficulty at work. Of the sample of 251 patients, 100% were valid cases. Below, we present the distance matrix of the variables analyzed (Table 3).

The cluster analysis showed the following: in the first stage, fatigue was clustered with work difficulty. In the second, anxiety was clustered with depression. In the third stage, sleep quality was clustered with rigidity. In the fourth stage, fatigue, work difficulty, sleep quality and stiffness were clustered. In the fifth, pain was combined with fatigue, work difficulty, sleep quality and stiffness. In the sixth, pain, fatigue, work difficulty, sleep quality and stiffness were combined with anxiety/depression (Table 4).

The dendrogram shows the grouping orders according to intensity: fatigue (FIQ_6) and work difficulty (FIQ_4) were the closest. Anxiety and depression were strongly linked and so on, as were sleep quality and stiffness (FIQ_7 and FIQ_8), (FIQ6 + FIQ4) with (FIQ_7 and FIQ_8) and this group with pain (FIQ_5) (Figure 2).

Cluster 1 consisted of patients with fatigue, work difficulty, sleep quality, stiffness and pain; however, a separate group with pain was identified in cluster 2 and anxiety and depression were linked in cluster 3. The interpretation of the combination scale indicates that the relationship between the variables anxiety and depression and functional disability (assessed using FIQ) was 25 times less relevant than the stronger relationship between fatigue and work difficulty or between anxiety and depression symptoms.

Cluster 3 analysis: The variables analyzed were: catastrophizing, surrender, mindfulness, acceptance, injustice, positive affect and negative affect. Of the sample of 251 patients, 96.8% were valid cases and 3.2% were missing values. In our study, a matrix of distances between the different psychological variables mentioned was performed (Table 5).

In addition, we present the average (inter-group) linkage of the aforementioned psychological domains (Table 6).

According to the clustering history under the criterion of proximity, they were grouped as follows: in the first stage, surrender and injustice are united in clusters. In the second stage, catastrophizing was united with surrender and injustice. In the third stage, acceptance and positive affect were clustered together. In the fourth stage, catastrophizing, surrender and injustice were clustered together with negative affect. In the fifth, mindfulness was joined with acceptance and positive affect, and, in the sixth, mindfulness, acceptance and positive affect were joined with catastrophizing, surrender, injustice and negative affect. The result of the previous process is represented by a graph called a dendrogram in the form of an inverted tree, where, on the vertical axis, the variables (and groupings) are placed and on the horizontal axis a relative measure of intensity is represented, rescaled inversely in distance.

The dendrogram shows the formation of the clusters, as well as the distances between them. It can be seen, for example, that the variables (observations) closest to each other are surrender and injustice, which form the first group (distance closest to 0). These two are joined with catastrophizing and then these three with negative affect, forming a relatively cohesive cluster. On the other hand, acceptance and positive affect were joined by proximity, and this cluster is joined in turn by mindfulness, forming the second cluster (Figure 3).

The observations that are most similar to each other in the first cluster are surrender, injustice and catastrophizing, and the last to be incorporated is negative affect. In the second cluster, the most similar and close variables are acceptance and positive affect and the most different is mindfulness. The two large clusters above are obviously part of a set of variables, which is the final total cluster (Figure 3).

### 3.3. Combination of Rescaled Distance Clusters

The following cluster classification was then obtained from the third cluster analysis: cluster 1: FM patients with high scores of surrender, unfairness, catastrophizing and negative affect; and cluster 2: FM patients with high scores of acceptance, positive affect and mindfulness.

## 4. Discussion

Our research has revealed that, by using cluster analysis on clinical data and psychological variables, we can identify different FM subgroups. In the first cluster analysis, which included clinical variables, anxiety and depression components and psychological domains, four subgroups were defined: cluster 1 displayed high scores in pain, fatigue, poor sleep quality, rigidity, anxiety/depression and difficulties at work; cluster 2 displayed high scores in injustice, catastrophizing, negative affect and positive affect; cluster 3 displayed high scores in mindfulness and acceptance; and, finally, cluster 4 displayed high scores in surrender. In the second cluster analysis, which included clinical variables and anxiety/depression components, three subgroups were defined: cluster 1 reported high fatigue scores, sleep quality, stiffness and difficulties at work; cluster 2 displayed high pain scores; and cluster 3 displayed high anxiety/depression scores. 

In the second cluster analysis, the objective of which was to group individuals into predefined subgroups and classify or organize them according to similar profiles with respect to the variables of pain, fatigue, sleep quality, stiffness, anxiety, depression and difficulty at work, fatigue, difficulty at work, sleep quality and stiffness appeared to be linked. It is important to highlight that difficulty at work does not appear associated or linked to pain (since it is the last one to be incorporated into the group, its distance to it being greater).

In the third cluster analysis that included psychological constructs, a first subgroup was defined, with high scores in surrender, injustice, catastrophizing and negative affect, and a second subgroup with high scores in acceptance, positive affect and mindfulness.

Our study has empirically demonstrated that the psychological and clinical factors studied could be partly responsible for both the explanation of the health outcomes of these FM patients and their chronification, supporting models that assume an interaction between biological, psychological and social factors [7].

A systematic review in the scientific literature identified different subgroups of FM according to criteria of patients without concomitant disease (FM type I), patients with rheumatic and autoimmune diseases (FM type II), with severe psychiatric disturbance (FM type III) and FM simulator patients (FM type IV) [30]. 

In this line, D’Souza et al. [31] identified two groups of FM patients, differentiated: one group by the degree of distress (anxiety and depression) and morning tiredness, and another group with psychological variables of catastrophism and perception of control. Similarly, Giesecke et al. [18] classified FM into different subgroups: the first, without symptoms of depression or anxiety, not catastrophic, with an internal locus of control and psychological factors not having a negative influence, which, in principle, leads to a better response to treatment; a second subgroup, with high levels of anxiety and depression, with an external locus of control, high level of pain, more years of disease progression and numerous symptoms, suffering significant negative social, cognitive and behavioral consequences, usually frequent medical visits and requiring multidisciplinary treatment; and a third subgroup, with severe pain, no anxiety or depression, their locus of control is internal, they are not catastrophic, they resolve or cope better with the situation than the previous ones and pharmacological treatment (analgesics, antidepressants and anticonvulsants) is more effective.

Our data agree with these studies, as they suggest the diagnostic heterogeneity that characterizes FM patients and that could be due to anxiety and depression symptomatology. These differences could be interpreted as being associated with one type of patients and not with others presenting with levels of pain, fatigue and joint stiffness [15].

Our study confirms the existence of clearly distinct patient profiles, which have also been described in previous studies [31,32]. However, the inclusion of various psychological variables highlights the importance of personalizing and optimizing pharmacological and psychological treatments; these results align with those of other reports [33,34].

Much of the multifactorial research on the biopsychosocial approach to FM shows significantly more psychological problems than healthy controls and also more than patients with chronic pain disorders, such as those with rheumatoid arthritis. Differences have been found in active coping strategies and, in the case of patients with FM, they present worse physical functioning, greater pain, poor coping strategies for their pain and a more limited support network than the healthy controls studied [35].

On the other hand, Fietta et al. [36] indicated that between 20–80% of FM patients present depressive disorders, according to DSM-IV and ICD diagnostic criteria. Other studies indicate that 47% of FM patients have an anxiety disorder and 50% present a depressive condition [14].

Studies have shown that there is a correlation between symptoms of psychopathology with the duration of the disease and pain [37]. Other findings studied the impact of FM on health-related quality of life and found that it was even greater than in other chronic diseases, such as rheumatoid arthritis or osteoarthritis [17].

In order to analyze the causes of the variability in depressive disorders, researchers have studied various patient subgroups. A study conducted by Broderick et al. [38] found that 92% of FM patients belonged to the dysfunctional group, 81% came from the interpersonal distress group and 39% from the adaptive group had severe depression scores. Later, the study by Verra et al. [39] replicated that study and found that the dysfunctional group had higher levels of depression than the interpersonal distress group and the adaptive group. The adaptative group had significantly lower levels of anxiety/depression, negative mood and catastrophizing compared to the other groups. 

There is great heterogeneity in relation to the presence of anxiety and depression disorders in the population affected with FM and although their prevalence is high, it cannot be established that these disorders are causal in nature or intrinsic to FM. However, there is scientific evidence that people with a higher level of anxiety and depression have a worse prognosis and their disease progresses negatively; it is also necessary to highlight a greater negative impact on their daily lives and a poorer quality of life. 

The subgroups of the study by Hasset et al. [40] was based on the affect balance style and used the PANAS questionnaire; therefore, four groups were obtained: (a) healthy with high positive affect/low negative affect; (b) low with low positive affect/low negative affect; (c) reactive with high positive affect/high negative affect; and (d) depressive with low positive affect/high negative affect. 

Boersma and Linton’s study [41] pointed out that because psychological factors are significantly associated with disability and pain in the FM patients, new therapeutic alternatives that include different psychological variables should be sought. Scientific research on FM and chronic pain has shown that cognitive-behavioral treatments have obtained good results and, when combined with pharmacological treatments and programmed physical exercise, greater effectiveness is achieved. 

This type of multidisciplinary intervention is usually applied in accredited hospital units and with people affected by FM with a degree of severity or severity much higher than that of patients with FM who are usually treated in primary care, the level of care where the present study was carried out. Given that there is scientific consensus on the high explanatory value of the psychosocial model in FM, and that the empirical data also support it, all this leads us to think of the importance of the application of cognitive and behavioral interventions. 

It is crucial to highlight the significance of introducing these multidimensional treatments for FM in primary care and making them accessible to all individuals with FM. This approach aims to enhance their overall health status, improve their quality of life and reduce the social and health burden and other costs derived from disability and sick leave in individuals with fibromyalgia.

### Study Limitations

To the best of our knowledge, only a few previous reports used cluster analysis to explore clinical and neuropsychological symptoms in the Spanish FM population. Moreover, we took into account the comorbid health conditions and clinical and neuropsychological domains in our analysis, allowing for a wide extension of our outcomes in routine clinical practice. However, several limitations need to be addressed. 

The main limitation is the cross-sectional study design, correlational in nature, where the type of sampling was performed in primary care; this limits the generalization of the results and may affect the risk of over-adjustment of the clusters found. Therefore, experimental and longitudinal study designs are more appropriate to better understand the differentiating profile of FM patients and their subsequent classification into various subgroups. However, the high effect sizes indicate that these results reflect the intergroup differences found. 

On the other hand, the potential to combine machine/deep learning to further enhance the performance and extend the application scenarios of the proposed methods is foreseen as a valuable tool for future studies [42,43]. 

## 5. Conclusions and Future Directions

The psychological (catastrophizing, surrender, acceptance, mindfulness, perceived injustice, positive affect and negative affect) and clinical (pain, fatigue, sleep quality, stiffness, anxiety, depression and difficulty or disability at work) factors studied are partially responsible for both the explanation for the health outcomes of these FM patients and their chronification. These empirical results support models that assume an interaction between biological, psychological and social factors. In the assessment of the FM patient, it is very important to use clinical variables, such as pain, fatigue, disability or difficulty working, sleep quality and stiffness, as well as other psychological variables related to anxiety, depression and other psychological constructs, in order to define the presence of subgroups and to consider the therapeutic approach in each of them.

## Figures and Tables

**Figure 1 biomedicines-11-02867-f001:**
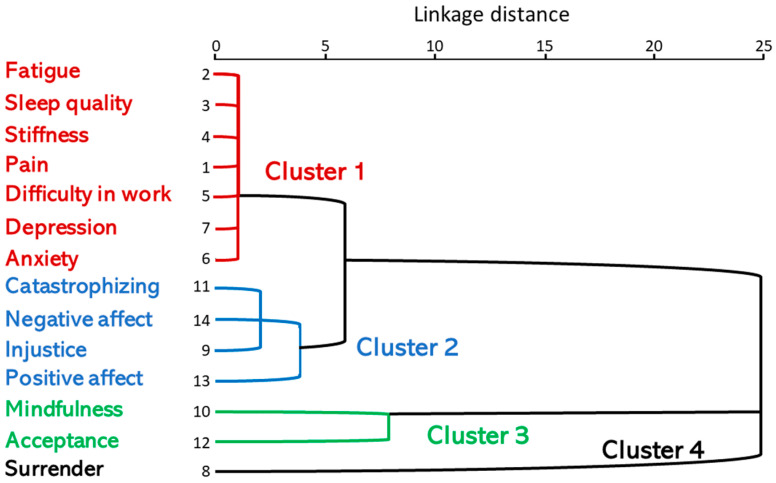
Dendrogram using a mean linkage between pain (FIQ_5); fatigue (FIQ_6); sleep quality (FIQ_7); stiffness (FIQ_8); anxiety (Total HADS-A); depression (Total HADS-D); difficulty at work (FIQ_4); surrender; injustice; catastrophizing (CATAST); negative affect (PANAS_negative); acceptance (CPAQ_TOTAL); positive affect (PANAS_positive); and mindfulness.

**Figure 2 biomedicines-11-02867-f002:**
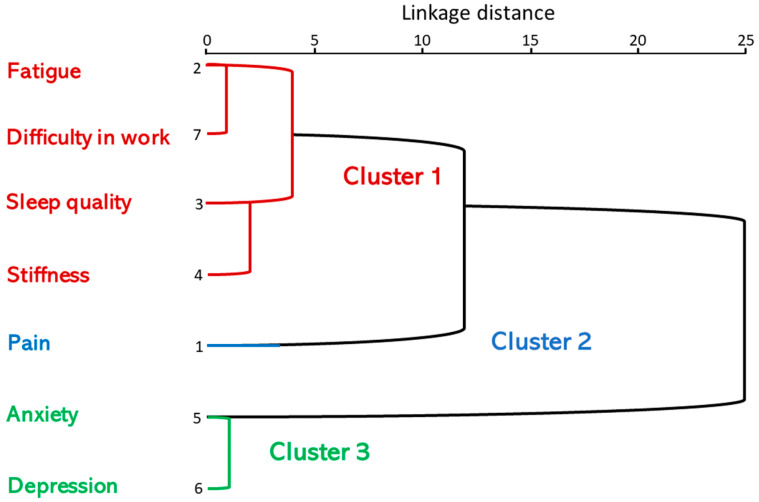
Dendrogram using a mean linkage between pain, sleep quality, fatigue, stiffness, anti-anxiety, depression and difficulty in work. Combination of rescaled distance clusters. The following cluster classification is then obtained from cluster analysis 2: cluster 1: FM patients with increased fatigue, sleep quality, stiffness and work difficulties scores; cluster 2: FM patients with pain; cluster 3: FM patients with raised anxiety/depression scores.

**Figure 3 biomedicines-11-02867-f003:**
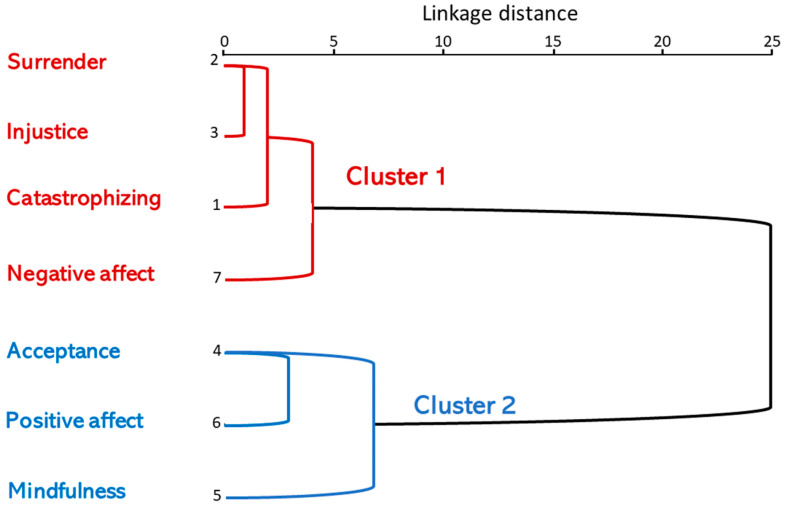
Dendrogram using an average linkage distance between surrender, injustice, catastrophizing, negative affect, acceptance, positive affect and mindfulness.

**Table 1 biomedicines-11-02867-t001:** Demographic and clinical characteristics of the study population (*n* = 251).

Variable	*n* (%) or Mean ± SD
Age (years)	52.4 ± 8
Gender (F/M)	241 (96.1)/10 (3.9)
Illness duration at diagnosis (years)	10.2 ± 9.3
Illness duration at inclusion (years)	18.3 ± 11.1
Current smoking	70 (27.9)
Ex-smoker (≥1 year)	50 (20)
Non-smoker	131 (52.2)
Marital status	
Married/living with partner	185 (73.7)
Separated/divorced	32 (12.8)
Single	23 (9.2)
Widower	11 (4.3)
Place of residence	
Zaragoza	221 (88.1)
Huesca	20 (7.9)
Teruel	10 (3.9)
Living arrangements	
Living with partner/spouse and children	119 (47.4)
Living with partner/spouse	80 (31.8)
Living alone	28 (11.2)
Living with other family	10 (3.9)
Others	14 (5.6)
Education	
Finished primary school	116 (46.2)
Finished secondary school	95 (37.8)
University graduate	32 (12.8)
No qualifications	8 (3.2)
Employment	
Employed	63 (25.1)
Disabled	53 (21.1)
Unemployed	38 (15.1)
Retired	34 (13.5)
Homemaker	32 (12.7)
Sick leave	31 (12.3)
Comorbid health conditions	
Chronic neck pain	238 (94.8)
Low back pain	230 (91.6)
Dry eyes	223 (88.8)
Anxiety/Depression	175 (69.7)
Cephalea/Migraine	193 (76.9)
Intestinal bowel syndrome	187 (74.5)
Menopause	169 (67.3)
Osteoarthritis	153 (60.9)
Rheumatoid arthritis	61 (24.3)

All values, except for age at inclusion and illness duration at diagnosis and at the time of study inclusion (as mean ± standard deviation (SD)), are displayed as numbers (percentages) of individuals.

**Table 2 biomedicines-11-02867-t002:** Cluster history. 1. Pain; 2. fatigue; 3. sleep quality; 4. rigidity; 5. difficulty of work; 6. anxiety; 7. depression; 8. surrender; 9. injustice; 10. mindfulness; 11. catastrophization; 12. acceptance; 13. positive affect; 14. negative affect.

Stage	Cluster * 1	Cluster * 2	Coefficients	Cluster ** 1	Cluster ** 2	Next Stage
1	2	3	886.000	0	0	3
2	1	5	972.250	0	0	4
3	2	4	1058.000	1	0	4
4	1	2	1560.958	2	3	5
5	1	7	4534.450	4	0	6
6	1	6	7649.042	5	0	10
7	11	14	31,097.000	0	0	8
8	9	11	36,317.500	0	7	9
9	9	13	77,097.667	8	0	10
10	1	9	108,588.036	6	9	12
11	10	12	151,021.000	0	0	12
12	1	10	513,915.114	10	11	13
13	1	8	526,653.865	12	0	0

* Cluster combined. ** Cluster stage appearing for the first time.

**Table 3 biomedicines-11-02867-t003:** Matrix of distances between pain, sleep quality, fatigue, stiffness, anxiety, depression and difficulty at work.

Variables	Pain	Fatigue	Sleep	Stiffness	Anxiety	Depression	Difficulty at Work
Pain	0.0000	206.240	283.696	232.052	329.424	314.157	191.962
Fatigue	206.240	0.0000	189.356	185.192	286.871	259.451	173.276
Sleep	283.696	189.356	0.0000	182.618	352.678	324.184	207.396
Stiffness	232.052	185.192	182.618	0.0000	312.836	279.140	182.064
Anxiety	329.424	286.871	352.678	312.836	0.0000	173.849	233.890
Depression	314.157	259.451	324.184	279.140	173.849	0.0000	227.812
Difficulty at work	191.962	173.276	207.396	182.064	233.890	227.812	0.0000

Note: 1 = pain (FIQ_5); 2 = fatigue (FIQ_6); 3 = sleep quality (FIQ_7); 4 = stiffness (FIQ_8); 5 = anxiety (Total HADS-A); 6 = depression (Total HADS-D); 7 = difficulty at work (FIQ_4).

**Table 4 biomedicines-11-02867-t004:** History of clusters of the variables pain (1), fatigue (2), sleep quality (3), rigidity (4), anxiety (5), depression (6) and difficulty at work (7).

Stage	Cluster * 1	Cluster * 2	Coefficients	Cluster ** 1	Cluster ** 2	Next Stage
1	2	7	173.276	0	0	4
2	5	6	173.849	0	0	6
3	3	4	182.618	0	0	4
4	2	3	191.002	1	3	5
5	1	2	228.488	0	4	6
6	1	5	292.044	5	2	0

* Cluster that is combined. ** Cluster appearing for the first time.

**Table 5 biomedicines-11-02867-t005:** Matrix of distances between the variables of catastrophizing, surrender, mindfulness, acceptance, injustice, positive affect and negative affect.

Variables	Catastrophizing	Surrender	Mindfulness	Acceptance	Injustice	Positive Affect	Negative Affect
Catastrophizing	0.0000	155.560	169.251	813.405	706.444	733.237	211.856
Surrender	155.560	0.0000	137.812	790.621	763.422	741.268	205.045
Mindfulness	169.251	137.812	0.0000	787.965	704.242	721.446	219.548
Acceptance	813.405	790.621	787.965	0.0000	298.573	192.085	742.781
Injustice	706.444	763.422	704.242	298.573	0.0000	278.596	703.264
Positive affect	733.237	741.268	721.446	192.085	278.596	0.0000	723.220
Negative affect	211.856	205.045	219.548	742.781	703.264	723.220	0.0000

**Table 6 biomedicines-11-02867-t006:** Clustering history. 1. Catastrophizing; 2. surrender; 3. perceived injustice; 4. acceptance; 5. mindfulness; 6. positive affect; 7. negative affect.

Stage	Cluster * 1	Cluster * 2	Coefficients	Cluster ** 1	Cluster ** 2	Next Stage
1	2	3	137.812	0	0	2
2	1	2	162.406	0	1	4
3	4	6	192.085	0	0	5
4	1	7	212.150	2	0	6
5	4	5	288.584	3	0	6
6	1	4	744.276	4	5	0

* Cluster that is combined. ** Cluster appearing for the first time.

## Data Availability

All data generated or analyzed during this study are included in this published article.

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
