# Peer review of "Hierarchical Cluster Analysis Based on Clinical and Neuropsychological Symptoms Reveals Distinct Subgroups in Fibromyalgia: A Population-Based Cohort Study"

_biomedicines, 2023, doi:10.3390/biomedicines11102867_

Round 1
Reviewer 1 Report
Thank you for submitting the manuscript. I read your article with great interest which is really well done.It is difficult to do research on fibromyalgia and it is difficult to read well-written works like yours.
However, I have to ask you for a couple of revisions to make your paper even more readable.In the introduction I would like you to underline how fibromyalgia is a pathology not only for adults but also for pediatric patients. It is necessary to make people understand how this debilitating pathology can start from childhood and profoundly affect people's psyche and life. In this regard I suggest you read and insert the following references: doi: 10.1002/art.22615. DOI: 10.3390/children9050637 doi: 10.1186/s12969-021-00529-x.
The second request I make of you is to have more demographic information about your sample, such as age, sex, work, and to insert it into a table.
For the rest, I enthusiastically congratulate you on your truly well-done article. I hope that my reviews are useful to you.
Kind Regards
Author Response
Dear Ms. Alexandra,
On behalf of the coauthors, I would like to thank you for giving us the opportunity to resubmit an updated version of the manuscript “Hierarchical cluster analysis based on clinical and neuropsychological symptoms reveals distinct subgroups in fibromyalgia: a population-based cohort study” for consideration in Biomedicines.
We appreciate the time and effort that the reviewer has taken to provide feedback on our manuscript and are extremely grateful for their comprehensive and insightful suggestions. The reviewer has raised new concerns, which we have carefully considered and made every effort to address. Accordingly, we have carefully included all their comments which are now highlighted through track changes in the manuscript.
Please find below our detailed point-by-point responses to the reviewer's comments and concerns, whereas the corresponding revisions are marked in red colored text throughout the manuscript. Additionally, we have carefully revised the manuscript to ensure that the text is optimally phrased and free from typos and/or grammatical errors. All page numbers refer to the revised manuscript file saved with tracked changes.
We believe this new edited version of the manuscript substantially improves the previous submission as a result of these revisions, and hope that this latest manuscript is now acceptable for publication in Biomedicines.
We would like to thank you for your consideration of our work and for inviting us to resubmit an updated version of the manuscript.
We look forward to hearing from you shortly.
Best regards,
Jesus Castro, Ph.D.
Reviewer #1
Thank you for submitting the manuscript. I read your article with great interest which is really well done. It is difficult to do research on FM and it is difficult to read well-written works like yours. However, I have to ask you for a couple of revisions to make your paper even more readable.
1- In the Introduction I would like you to underline how FM is a pathology not only for adults but also for pediatric patients. It is necessary to make people understand how this debilitating pathology can start from childhood and profoundly affect people's psycho- and life. In this regard, I suggest you read and insert the following references: doi: 10.1002/art.22615; doi: 10.3390/children9050637 & doi: 10.1186/s12969-021-00529-x.
We agree with the reviewer's comments. We have inserted the refs requested by the reviewer (see refs. 10, 11, & 12 in the updated version of the manuscript).
2- I make of you to have more demographic info- about your sample, such as age, sex, and work disability, and to insert it into a table
Thanks to the reviewer for this point. We have now inserted information corresponding to the demographic and clinical features of the study population (see Table 1 on page 5 in the updated manuscript).
Reviewer 2 Report
This manuscript used a hierarchical cluster analysis based on clinical and neuropsychological symptoms to reveal distinct subgroups in individuals with FM, the topic looks interesting. My comments are as follows:
1) English writing improvement is suggested.
2) Both motivations and contributions are unclear in Abstract and Introduction, please refine them.
3) Separate related work section should be considered by the authors to review state-of-the-arts should in this area.
4) High-quality figures are strongly suggested to better demonstrate both the proposed method and experimental results.
5) The experiments are not sufficient, more baseline methods, scenarios, data and evaluation metrics should be included to support the proposed method.
6) The authors should comprehensively discuss both the limitations of the proposed method and future directions. For example, the potential to combine machine/deep learning to further enhance the performance and extend the application scenarios of the proposed method. Some related papers are recommended, which are better included in the reference list: physics-informed deep learning for musculoskeletal modeling: predicting muscle forces and joint kinematics from surface emg, IEEE TNSRE, and online spatiotemporal modeling for robust and lightweight device-free localization in nonstationary environments, IEEE TII.
English writing enhancement is suggested.
Author Response
Dear Ms. Alexandra,
On behalf of the coauthors, I would like to thank you for giving us the opportunity to resubmit an updated version of the manuscript “Hierarchical cluster analysis based on clinical and neuropsychological symptoms reveals distinct subgroups in fibromyalgia: a population-based cohort study” for consideration in Biomedicines.
We appreciate the time and effort that the reviewer has taken to provide feedback on our manuscript and are extremely grateful for their comprehensive and insightful suggestions. The reviewer has raised new concerns, which we have carefully considered and made every effort to address. Accordingly, we have carefully included all their comments which are now highlighted through track changes in the manuscript.
Please find below our detailed point-by-point responses to the reviewer's comments and concerns, whereas the corresponding revisions are marked in red colored text throughout the manuscript. Additionally, we have carefully revised the manuscript to ensure that the text is optimally phrased and free from typos and/or grammatical errors. All page numbers refer to the revised manuscript file saved with tracked changes.
We believe this new edited version of the manuscript substantially improves the previous submission as a result of these revisions, and hope that this latest manuscript is now acceptable for publication in Biomedicines.
We would like to thank you for your consideration of our work and for inviting us to resubmit an updated version of the manuscript.
We look forward to hearing from you shortly.
Best regards,
Jesus Castro, Ph.D.
Reviewer #2:
This manuscript used a hierarchical cluster analysis based on clinical and neuropsychological symptoms to reveal distinct subgroups in FM, the topic looks interesting. My comments are as follows:
1- English writing improvement is suggested.
Done.
2- Both motivations and contributions are unclear in the Abstract and Introduction, please refine them.
Done.
3- A separate related work section should be considered by the authors to review the state-of-the-art in this area.
Done.
4- High-quality figures are strongly suggested to better demonstrate both the proposed method and experimental results.
Done
5- The experiments are not sufficient, more baseline methods, scenarios, data, and evaluation metrics should be included to support the proposed methods.
Done
6- The authors should comprehensively discuss both the limitations of the proposed method and future directions. For example, the potential to combine machine/deep learning to further enhance the performance and extend the application scenarios of the proposed method. Some related papers are recommended, which are better included in the reference list: Physics-informed deep learning for musculoskeletal modeling: predicting muscle forces and joint kinematics from surface EMG (https://www.researchgate.net/publication/361756272), and Online spatio-temporal modeling for robust and lightweight device-free localization in non-stationary environments (IEEE Transactions on Industrial Informatics 2023;19:8528-38), doi: https://doi.org/10.1109/TII.2022.3218666).
The suggested references have been now added (see 42 and 43 in the updated version of the manuscript).
Round 2
Reviewer 1 Report
Thank you for submitting the new version of the manuscript. I am extremely satisfied. Congratulations.
Kind Regards
Reviewer 2 Report
The authors have addressed the issues, I have no more comments.
No.